# Comprehensive Genomic Characterization of Fifteen Early-Onset Lynch-Like Syndrome Colorectal Cancers

**DOI:** 10.3390/cancers13061259

**Published:** 2021-03-12

**Authors:** Mariano Golubicki, Marcos Díaz-Gay, Laia Bonjoch, Sebastià Franch-Expósito, Jenifer Muñoz, Miriam Cuatrecasas, Teresa Ocaña, Soledad Iseas, Guillermo Mendez, Marcela Carballido, Juan Robbio, Daniel Cisterna, Enrique Roca, Antoni Castells, Francesc Balaguer, Sergi Castellví-Bel, Marina Antelo

**Affiliations:** 1Oncology Section, Hospital of Gastroenterology “Dr. C. B. Udaondo”, Buenos Aires C1264, Argentina; mariano.golubicki@gmail.com (M.G.); soledad.iseas@gmail.com (S.I.); mendezdoc@hotmail.com (G.M.); mcarballido@gmail.com (M.C.); robbiojuan@gmail.com (J.R.); enlroca@yahoo.com.ar (E.R.); 2Molecular Biology Lab, Hospital of Gastroenterology “Dr. C. B. Udaondo”, Buenos Aires C1264, Argentina; dancis99@yahoo.com; 3Centro de Investigación Biomédica en Red de Enfermedades Hepáticas y Digestivas (CIBERehd), Gastroenterology Department, Institut d’ Investigacions BiomèdiquesAugust Pi i Sunyer (IDIBAPS), Hospital Clínic, 08036 Barcelona, Spain; mdiazgay@health.ucsd.edu (M.D.-G.); bonjoch@clinic.cat (L.B.); franches@mskcc.org (S.F.-E.); jenifer.munoz@ciberehd.org (J.M.); mocana@clinic.cat (T.O.); castells@clinic.cat (A.C.); fprunes@clinic.cat (F.B.); 4Department of Cellular and Molecular Medicine, University of California, San Diego, La Jolla, CA 92093, USA; 5Pathology Department, Hospital Clínic, 08036 Barcelona, Spain; mcuatrec@clinic.cat

**Keywords:** colorectal cancer, biallelic somatic alteration, early-onset cancer, Lynch-like syndrome, mismatch repair, whole-exome sequencing

## Abstract

**Simple Summary:**

The most prevalent type of hereditary colorectal cancer is called Lynch syndrome and it is characterized by a tumor phenotype called microsatellite instability (MSI). This disease is a consequence of germline (inheritable) variants in any of the four mismatch repair (MMR) DNA genes, being their identification essential to ensure their appropriate diagnosis and implementation of preventive measurements. Nevertheless, only 50% of patients with MSI and suspected Lynch syndrome actually carry a germline pathogenic variant in an MMR gene that explains the clinical entity. The remaining 50% are termed Lynch-like syndrome, and their causes remain unknown. In this work, we tried to elucidate the molecular mechanisms that underlie this rare entity in a group of early-onset Lynch-like syndrome colorectal cancer, through whole-exome sequencing of germline and tumor samples. We observed that one-third of these patients have somatic alterations in genes associated with the MMR system and that these could be the mechanism causing their unexplained MSI. Furthermore, we found that patients who showed biallelic somatic alterations also carried germline variants in new candidate genes associated with DNA repair functions and that this could be, partly, the cause of the early onset in this cohort.

**Abstract:**

Lynch-like syndrome (LLS) is an increasingly common clinical challenge with an underlying molecular basis mostly unknown. To shed light onto it, we focused on a very young LLS early-onset colorectal cancer (CRC) cohort (diagnosis ≤ 40 y.o.), performing germline and tumor whole-exome sequencing (WES) of 15 patients, and additionally analyzing their corresponding tumor mutational burden (TMB) and mutational signatures. We identified four cases (27%) with double somatic putative variants in mismatch repair (MMR) core genes, as well as three additional cases (20%) with double *MSH3* somatic alterations in tumors with unexplained MSH2/MSH6 loss of expression, and two cases (13%) with *POLD1* potential biallelic alterations. Average TMB was significantly higher for LLS cases with double somatic alterations. Lastly, nine predicted deleterious variants in genes involved in the DNA repair functions and/or previously associated with CRC were found in nine probands, four of which also showed MMR biallelic somatic inactivation. In conclusion, we contribute new insights into LLS CRC, postulating *MSH3* and *POLD1* double somatic alterations as an underlying cause of a microsatellite instability (MSI) phenotype, proposing intrinsic biological differences between LLS with and without somatic alterations, and suggesting new predisposing candidate genes in this scenario.

## 1. Introduction

Colorectal cancer (CRC) is the second most commonly diagnosed cancer and the third leading cause of cancer deaths in most developed countries, with a mean age of 70 years at diagnosis [1]. However, up to 15% of all cases occur before the age of 50 years [2], and recent epidemiological studies suggest that incidence and mortality of these early-onset CRCs are increasing [3,4]. Several hereditary (Lynch syndrome, familial adenomatous polyposis, *MUTYH*-associated polyposis) and non-hereditary diseases (ulcerative colitis, Crohn’s disease) are associated with early-onset CRC, and their early detection is essential because of the increased lifetime CRC risk and the potential positive impact of preventive measures on survival [2]. Inflammatory bowel disease and colonic polyposis syndromes are easily identified from their phenotypic features, but Lynch syndrome patients do not have a characteristic clinical phenotype and are often missed, especially in the absence of a family history of cancer.

Lynch syndrome (LS) is the most common hereditary cancer syndrome, affecting an estimated 1 in 300 individuals and have reported to have increased incidences of cancers in the colon, rectum, endometrium, ovaries, stomach, small bowel, bile duct, pancreas, and upper urinary tract. LS is also the most common hereditary cause of CRC, accounting for approximately 1–4% of all cases [5]. It is an autosomal dominant condition caused by germline pathogenic or likely pathogenic variants in a DNA mismatch repair (MMR) gene. *MSH2* and *MLH1* account for most Lynch syndrome-associated CRCs. This syndrome has a marked genetic-dependent variable penetrance for CRC and endometrial carcinoma (30–80%), and patients are at increased risk for other extra-colonic tumors. Annual surveillance colonoscopies and total hysterectomy reduce cancer mortality [6]. Additionally, the identification of a causal deleterious variant in one of the MMR genes leads to genetic pre-symptomatic diagnosis in relatives, focusing screening measures on variant carriers.

While the hallmark of this disease is tumor MMR deficiency (MMRd), defined by the presence of microsatellite instability (MSI) and/or absence of MMR protein expression by immunohistochemistry (IHC), BRAF V600E wild type, and no *MLH1* promoter hypermethylation, a diagnosis of Lynch syndrome requires the presence of a deleterious germline variant in a DNA MMR gene [7]. MMRd tumors without a germline variant in any of the four MMR genes may account for as many as 70% of cases [8]. These cases are termed “Lynch-like syndrome” (LLS), and management decisions in these patients and their families are complicated because of unconfirmed suspicions of hereditary cancer [9]. In addition to the hypothesis regarding possible “cryptic” deleterious germline variants in MMR genes, there are two other possible explanations proposed: unknown germline pathogenic or likely pathogenic variants affecting other than MMR genes may drive the tumor towards MSI (as it has been recently described for *MUTYH* and *POLE* genes in isolated LLS cases [10,11,12]), and biallelic somatic inactivation of an MMR gene in LLS tumors may cause MMRd (as it has been suggested in several reports [13,14,15,16]).

Besides the MMR genes, the wider LLS tumors’ somatic alteration profiles are unfamiliar, and there is actually scant information about underlying somatic characteristics and the complex processes behind the MMR protein inactivation. To shed light onto it, we thus focused on a very young LLS early-onset CRC cohort to elucidate the underlying molecular basis of this increasingly common clinical challenge, by performing germline and tumor whole-exome sequencing (WES) of 15 patients with early-onset LLS CRC.

## 2. Results

### 2.1. Patients

Fifteen patients with CRC LLS diagnosed ≤40 y.o. were included in this study. Demographic, clinic-histopathological, and MMR characterization of this cohort is presented in Table 1. Mean age at diagnosis was 30.2 years (SD 7.2); 9 (60%) were women, and 11 (73%) patients had tumors located distal to the splenic flexure. The majority of cases (12; 80%) were diagnosed at advanced stages (III-IV). Lack of MLH1/PMS2 expression was as frequent as MSH2/MSH6 (both 7/15; 47% each). Isolated loss of MLH1 and MSH6 (2/15; 0.7% each) were less frequent. All patients fulfilled Bethesda 1 criteria, and half of them also presented MSI-suggestive histology (Bethesda 3 criteria).

### 2.2. Somatic Biallelic Alterations in MMR Core and MMR-Associated Genes

When exploring somatic alterations in the MMR core genes (*MLH1*, *MSH2*, *MSH6*, *PMS2*), four of 15 patients (27%) had two somatic heterozygous variants explaining their IHC findings, one in *MSH2* (LLS18) and three in *MLH1* (LLS17, LLS21, LLS9045). Seven of these variants were truncating and potentially pathogenic; although the remaining one was a missense variant (p.Gly67Arg), Clinvar and three in silico prediction tools classified it as pathogenic.

On the other hand, when MMR-associated genes (*MSH3*, *MLH3*, *PMS1*, *MUTYH*, *POLE*, *POLD1*) were also considered, we observed five additional cases (33%) with putative biallelic alterations. Three patients with IHC MSH2/MSH6 loss of expression (LLS02, LLS09, LLS10) showed one somatic *MSH3* truncating variant and associated LOH (considered as a second hit). Two additional patients were found to carry biallelic *POLD1* potentially pathogenic variants with MSH2/MSH6 and MLH1/PMS2 loss of expression, respectively (LLS9049, LLS5159). Of note, LLS10 showed concomitant *MSH3* and *POLD1* biallelic alterations.

Lastly, LLS9045 (which showed *MLH1* biallelic somatic variants), also presented biallelic alterations in *MLH3* (a missense variant with associated LOH).

In the remaining six patients (40%), no biallelic somatic alterations were identified in the analyzed genes. Of these, 3 showed MLH1/PMS2 loss of expression (LLS15, LLS5604), one isolated MSH6 deficiency (LLS06), and two MSH2/MSH6 (LLS15, LLS20).

All results are presented in Figure 1 and all somatic findings are summarized in Table 2. The InSiGHT database classifications (http://www.insight-database.org/classifications/ access date: 16 January 2021) for all MMR variants are shown in Appendix A. Additionally, all somatic variants classified according to the ACMG criteria as pathogenic, likely pathogenic, uncertain significance, likely benign and benign, are shown in Appendix A.

### 2.3. Somatic Monoallelic Alterations in MMR Core and MMR-Associated Genes

We then evaluated monoallelic variants. In patients with no potential biallelic inactivation of MMR core and MMR associated gen, only LLS19, with MSH2 loss of IHC expression, showed single *MSH2* and *MSH6* variants, with no associated LOH.

On the other hand, additional monoallelic variants were identified in patients with biallelic inactivation, and are summarized in Figure 1 and Table 2.

### 2.4. Somatic Alterations in Additional Cancer Genes

Additional data on the most common CRC mutated genes are reported in Appendix A. *APC* and *FBXW7* were mutated in 7/15 cases (47%), *ATM* and *KRAS* in 6/15 (40%), and *TP53* and *PTEN* showed variants in 2/15 (13,3%). One single patient presented one variant in *SMAD4*, and no variants were found in *NRAS*.

### 2.5. MSH3 and MSI Association

According to our results, we hypothesized that *MSH3* biallelic somatic inactivation could be involved in LLS syndrome associated CRC. We therefore investigated if *MSH3* variants could be associated with MSI in 3806 CRC cases with somatic profiling data from cBioPortal (http://www.cbioportal.org access date: 16 January 2021). Although we observed 67 CRCs with *MSH3* variants, only 13 (13/3806, 0.34%) had at least one truncating *MSH3* somatic pathogenic variant with no associated variants in *MLH1*, *MSH2*, *MSH6*, or *PMS2*, nor *MLH1* silencing by hypermethylation. Among these 13 cases, 85% (11/13) were MSI, and although one case had no MSI data, the number of variants was similar to those displaying MSI (sample ID: coadread_dfci_2016_60 and 1,937 variants). The remaining case (1/13, 8%) was reported as microsatellite stable (MSS) (Appendix A).

### 2.6. Tumor Mutational Burden (TMB)

Five out of 15 patients (33.3%) showed mutational burdens compatible with ultra-hypermutation phenotype, accounting for over 100 mutations per megabase, and four of them (4/5, 80%) were patients with biallelic inactivation. Seven additional patients displayed a hypermutated phenotype with 10–100 mutations per megabase, and the remaining three were non-hypermutated, with less than 10 mutations per megabase (Table 1 and Figure 1).

Searching for molecular differences between patients with double somatic alterations in MMR genes, *MSH3* or *POLD1*, and those with no somatic inactivation, we measured TMB and compared it according to these two groups. Interestingly, mean TMB of the first group was significantly higher (100.8 versus 17.0 mutations/Mb; Wilcoxon rank-sum test; *p*-value = 0.002). For this analysis, we excluded the second group LLS19 sample, which presented a particular mutational signature related to the patient’s history of previous temozolomide treatment (see Section 2.7).

We then compared TMB according to tumor location, observing that patients with tumors distal to splenic flexure presented higher mutations per megabase than those proximal to splenic flexure (88.3 versus 8.5 mutations/Mb; Wilcoxon rank-sum test; *p*-value = 0.0019) (Appendix A).

### 2.7. Mutational Signatures

Further somatic characterization was performed by analyzing mutational signatures, considering single base substitutions (SBS) and small insertion and deletions (ID), in order to have a closer look at the different mutational mechanisms implicated in our cohort. De novo extraction considering SNVs revealed six active mutational signatures, subsequently decomposed into nine previously indexed in version 3.1 of COSMIC database reference set (Jun 2020) (Appendix A) [17,18].

One of the previously known mutational signatures found in our cohort was SBS11, associated with temozolomide (TMZ) chemotherapy treatment. This signature was found as the major contributor in the LLS19 patient´s CRC, who reported a previous TMZ treatment for a non-Hodgkin lymphoma diagnosed at the age of 21 years, four years prior to the MSI CRC diagnosis.

Besides this particularity, and as expected, the rest of the mutational signature contributions in the cohort were dominated by those related to MMR deficiency (45.1% on average), with up to four different signatures present (SBS6, SBS15, SBS20, and SBS21). Interestingly, all nine tumors harboring somatic biallelic inactivation in MMR, MMR associated, or hypermutation genes showed a significantly higher number of mutations linked to MMR deficiency signatures versus those with monoallelic or no somatic mutations (1795.9 vs. 206.8 mean mutations; Wilcoxon sum-rank test; *p*-value = 0.004; excluding LLS19 case).

In the case of small insertions and deletions (indels), four different mutational signatures were extracted and decomposed, corresponding to three known COSMIC v3.1 signatures related to MMR deficiency (ID1, ID2, and ID7), and one new signature that had not been previously reported (referred to as IDA) (Appendix A). This new signature is characterized by de novo insertions of 5 or more nucleotides. Interestingly, signature IDA was found active in four out of the 5 tumors without biallelic inactivation, whereas only in one of the inactivated samples (11%). Indeed, the indel spectrum of the only case presenting a MSS tumor (and isolated loss of MSH6 by IHC), LLS06, is exclusively composed by IDA-linked alterations.

### 2.8. Germline Candidate Genes

In order to explore the hypothesis that somatic inactivation of MMR system could be a random consequence of deleterious germline variants in unknown genes, we analyzed dominant candidates and chose those with specific DNA repair functions from our WES data. A recessive candidate from this cohort was recently published [19].

All germline DNA samples were sequenced with good quality, with a mean coverage higher than 60×. This prioritization strategy yielded 10 very rare predicted deleterious variants in 9 DNA repair associated genes (gnomAD < 0.1% or not present), and all of them scored over 20 in CADD Phred score. Results are summarized in Table 3.

Ten patients (66.7%) carried predicted deleterious variants in 9 genes with functions involved in DNA repair mechanism; five patients without somatic biallelic variants carried five of these variants: four missense variants in *ERCC6*, *POLE*, *EXO5*, and *RAD52*, and one in a donor splice site in *RECQL4*. The remaining five variants were identified in patients with biallelic somatic inactivation: four missense in *PALB2*, *UVRAG*, *RAD54L,* and *MCM2*, and one in the acceptor splice site in *RAD54L*. Nine variants were cataloged as a VUS (variant of uncertain significance) by ACMG/AMP criteria, and one in *RECQL4*(c.1878+1G > C) was scored as pathogenic, disrupting the donor splice site of the exon 11. Results are summarized in Table 3.

The most promising candidate genes were *POLE*, *ERCC6*, *RAD54L*, and *PALB2*. The *POLE* variant (c.1847G > A; p.Arg616His) identified in LLS20 was outside the exonuclease domain, and its respective tumor was not hypermutated and did not present any mutational signatures previously associated with *POLE* variants. The *ERCC6* variant (c.1670G > A, p.Arg557His) affecting the helicase ATP-binding domain was present in LLS06, whose tumor was MSS, showed low TMB, and no contribution of MMR deficiency-associated ID mutational signatures (Figure 1).

Two different variants (c.767-2A > G and c.17C > T, p.Ala6Val) in *RAD54L* were identified in two patients (LLS9 and LLS10, respectively); the first variant impacted the helicase ATP-binding domain, and the second one, which was a missense variant, impaired the region of interest related to chromatin remodeling and ATPase activity. Interestingly, both patients had somatic inactivation of *MSH3* with associated LOH, and similar TMB and mutational signature profiles.

Lastly, the *PALB2* missense variant (c.2606C > T;p.Ser869Phe) was present in LLS18, whose respective tumor showed the largest indel burden. This was the only candidate variant present in Clinvar (as a VUS), and previously reported in a suspected Lynch patient [20].

## 3. Discussion

In this study of a well-annotated cohort of 15 early-onset CRC LLS patients, we performed germline and tumor WES analysis to elucidate the underlying molecular basis of an increasingly common clinical challenge. We identified four cases (27%) with double somatic alterations in MMR core genes explaining their findings on IHC testing, as well as three additional cases with double *MSH3* somatic alterations in tumors with unexplained MSH2/MSH6 loss of expression, and two cases with potential *POLD1* biallelic variants. These results support the idea that *MSH3* testing may be considered in CRC samples with unexplained MSH2 loss of IHC expression.

In approximately 30% of CRC cases [8,21], the presence of MSI or loss of immunochemical expression of MMR genes is found with no corresponding germline pathogenic variants. These LLS patients [9] are clinically represented by at least two different subsets. The first group includes cases in which the clinical characteristics, such as early onset of CRC or a strong family cancer history, suggest a hereditary origin, whereas the second subset includes a significant proportion of LLS families who experience late CRC onset and have no family history of cancer, and therefore, may represent sporadic tumors with no need for preventive measures for themselves or their relatives. LLS appears to be a heterogeneous condition between these two situations.

Testing for somatic variants in MMR genes has been proposed for differential diagnosis between hereditary and sporadic cases in this scenario, and some groups have proposed investigating somatic variants in MMR genes that might explain sporadic CRC cases with LLS [22]. We identified six studies, summarized in Appendix A, that have specifically analyzed combinations of biallelic somatic MMR gene variants or single somatic MMR gene variants with associated LOH in patients with CRC and LLS. Sourouille et al. [13], Mensenkamp et al. [14], Geurts-Giele et al. [15], Haraldsdottir et al. [16], and more recently Xicola et al. [23] and Porkka et al. [24], have identified 24% (4/17), 48% (11/23), 51% (17/33), 93% (14/15), 54% (6/11), and 79% (11/14) cases with MMR double somatic alterations, respectively. The 27% (4/15) rate of cases in our study falls within the range of these previous reports and with a similar number of analyzed patients. This very broad range (24% to 93%) of MMR double somatic alterations in CRC LLS patients probably reflects underlying clinical differences between the described cohorts, such as the clear variation in age at CRC diagnosis, ranging from 30 to 65. Of note, Sourouille et al. [13], with a 24% rate of double somatic alterations, did not analyze LOH in samples with one MMR somatic variant and is likely missing some cases. Therefore, our cohort could have the lowest percentage of cases with double MMR somatic alterations, coherent with the fact that it presents the youngest age range at CRC diagnosis. Hence, an interesting hypothesis is that the older the age at CRC diagnosis, the higher the probability of double somatic MMR alteration, and the lower the probability of a hereditary origin.

*POLE*, *POLD1*, *MUTYH, MSH3*, *MLH3*, and *PMS1* somaticalterations analysis in our eleven unexplained CRC LLS revealed some interesting findings. We observed double somatic alterations in five of these cases, two presenting biallelic *POLD1* variants, and three showing double *MSH3* somatic alterations.

MSH3 is part of a MutS homolog heterodimer with MSH2, named MutSb, which partially overlaps with MutSa, a heterodimer composed of MSH2 and MSH6 with mispair-recognition specificity [25]. In mouse models, elimination of either MSH6 or MSH3 alone still maintains some functional MMR activity, which is consistent with the persistence of the MutSa or MutSb heterodimer, respectively. In humans, MutSa effectively binds single-base substitution and small (single-base) indel mispairs, while MutSb has a solid affinity for larger indel loops with up to ten unpaired bases [26,27]. Thus, inactivation of MSH3 in human cells not only results in MSI that hold dinucleotide repeats but also results in MSI at certain loci with tetranucleotides repeats, termed EMAST [28]. Furthermore, mouse *Msh3* deficiency leads to a partial MMR defect and MSI [29,30].

In our study, we found three patients that showed somatic MSH3 truncating variants associated with LOH, in MSH2-/MSH6- tumors. Although we hypothesize that the MSH3 alterations could be the underlying cause of the MMR inactivation, we cannot fully discard the contrary. Indeed, the mutational signatures associated with these tumors are mostly related to an MMR deficiency. We did not search for EMAST features and, therefore, we cannot differentiate between mono/dinucleotides alterations (associated with MMR deficiency) or polynucleotides alterations (associated with EMAST). However, it is well known that MMR-deficient tumors also shared elevated microsatellite alterations at selected tetranucleotide repeats (EMAST) in previous studies [31]. On the other hand, no somatic mutations or LOH were found in MSH2 in these patients, and only two *MSH6* VUS variants were detected (in LLS09 and LLS10). To our knowledge, a specific mutational signature related to MSH3 deficiency is not yet known, besides the previous evidence that MSH3 deficiency causes EMAST or EMAST with low levels of MSI at loci with dinucleotide repeats in CRC. The COSMIC signatures used in our study do not differentiate between mono or polynucleotide repeats. However, it is likely that the *MSH3* mutational signature shares some features with those currently known in the COSMIC database to be associated with MMR defects. LLS09 and LLS10 showed concomitant MSH3/MSH6 and MSH3/POLD1 biallelic alterations, respectively. The LOH observed in MSH3 argues in favor that they occurred prior to those in MSH6 and POLD1. However, this matter should be taken very cautiously and further investigated in order to draw a more solid conclusion.

Regarding germline CRC predisposition, common *MSH3* polymorphisms were significantly associated with CRC and prostate cancer as low-penetrance risk alleles [32,33,34], and recently high-penetrance pathogenic biallelic germline *MSH3* variants have also been associated with CRC and colorectal polyposis syndromes [35]. Therefore, several lines of evidence support the causal relevance of MSH3 deficiency in the initiation of genetic instability and tumorigenesis.

Finally, CBioPortal findings may represent a means of externally validating our *MSH3* observations. We postulate that double *MSH3* somatic alterations may represent a different underlying mechanism for the generation of MSI, with loss of MSH2/MSH6 expression by IHC, at least in CRC. Almost half of the 34 unexplained MMR deficient tumors from the compared cohorts (Appendix A) presented MSH2/MSH6 protein loss of expression by IHC; it would be interesting to search for *MSH3* somatic alterations in these cohorts to support our findings.

The *POLD1* gene has been widely associated with MMR deficient tumors [11] but the evidence is not clear about if there is a direct relation between somatic loss of function *POLD1* variants and deficiency of MMR proteins or it is a consequence of a mutator phenotype that impact the MMR genes. In our study, we found that it may be possible that loss of function *POLD1* variants associated with LOH could impair the expression of MMR proteins, which could be the case in LLS9045 and LLS5159 patients. Moreover, these patients also showed mutational signatures associated with *POLD1* defects in combination with MMR deficiency signatures but no variants in MMR core genes. Interestingly, LLS5159 had the same variant reported as a germline in a recent study [36] and that patient’s tumor also showed hypermutation phenotype but was MSS. The biallelic inactivation of *POLD1* in LLS5159 could result in this case in an MSI phenotype.

Average TMB was significantly higher for LLS cases with double somatic alterations compared to LLS cases without double somatic alterations in the explored genes (*MMR*, *POLD1,* and *MSH3*) and, surprisingly for tumors left-sided, on the contrary to the reported [37,38]. This fact could have potential clinical implications, such as hypermutability and neoantigen-induced immunoreactions, rendering MSI tumors candidates for PD-1 blockade-based immunotherapy [39]. Le, D.T. et al. reported that non-Lynch patients with MMR-deficient tumors responded significantly better than Lynch syndrome patients, but no data are available for LLS tumors specifically. Although the presence of certain particularly immunogenic neoantigens, rather than the TMB, may ultimately determine the response of LLS tumors to PD-1 blockade [40], it would be interesting to explore if differences in tumors with particular somatic alterations reflect different levels of PD-1 blockade responses among patients with CRC LLS.

Almost all patients (14/15), except LLS35604, showed at least one mutational signature related to MMR deficiency. The LLS19 patient had a non-Hodgkin lymphoma four years before her CRC, which showed a major SBS11 contribution. This signature is associated with alkylating agents and therefore suggests that her CRC was a consequence of her previous temozolomide treatment. Our findings are in part in concordance with recent reports of high frequency double somatic alterations in MMR genes among patients who underwent treatment with temozolomide [41]. We believe that in the LLS scenario, the mutational signature approach may be useful in patients with prior malignancies, by revealing if the current CRC is a potential consequence of previous chemotherapy treatments [42].

To continue our analysis we explored the presence of germline pathogenic variants in genes involved in the DNA repair functions. We found 9 candidate genes in 10 patients, of which only *POLE* has been previously associated with early-onset MSI CRC. *ERCC6* and *PALB2* have also been related to CRC predisposition but not specifically with LLS [5,11,43]. Although the remaining six candidates have not been linked to CRC predisposition, they have been related to cancer development and genome maintenance [44,45,46,47,48,49,50,51]. In any case, further functional studies and replication in additional cohorts will be needed in order to further confirm the identified potential candidates for LLS germline predisposition.

## 4. Materials and Methods

### 4.1. Patients

Fifteen unrelated patients from two institutions (Hospital Udaondo, Buenos Aires, Argentina, and Hospital Clínic, Barcelona, Spain) with CRC diagnosed ≤40 y.o. and LLS were included in this study. Demographic and clinical and pathological characteristics were obtained from patients’ medical records, and family history of cancer in first and second-degree relatives was collected by a personal interview. All patients presented tumors with MSI and/or IHC loss of MSH2, MSH6, PMS2, or loss of MLH1 with somatic wild-type V600EBRAF and no *MLH1* methylation, as well as not detected germline pathogenic variants in the MMR genes or *EPCAM*. Relevant demographic and clinical and pathological features are presented in Table 1. It should be noted that patient LLS06 presented absence of the MSH6 protein expression and MSS in her tumor.

CRC tissue samples and germline DNA were obtained for each patient from the respective institution biobank. Matched tumor and germline DNA samples were used to perform WES when available (15/15 patients) with optimal quantity and quality. A QIAamp Tissue Kit (Qiagen, Redwood City, CA, USA) was used to isolate tumor DNA from formalin-fixed paraffin-embedded (FFPE) tissue following the manufacturer’s instructions, achieving 70–80% tumor cells among all 15 available samples.

This research has been approved by the Institutional Review Board and ethics committee on 22 December 2015. Patients signed a protocol-specific informed consent.

### 4.2. Whole Exome Sequencing

WES was performed in tumor and germline samples of selected patients using the HiSeq2000 platform (Illumina, San Diego, CA, USA) and SureSelectXT Human All Exon v5 kit (Agilent, Santa Clara, CA, USA) for exon enrichment. Indexed libraries were pooled and massively parallel-sequenced using a paired-end 2 × 75 bp read length protocol. Quality control of sequencing data was performed in all samples prior to their analysis using the Real-Time Analysis software sequence pipeline (Illumina). Additionally, the proportion of all shared exome regions sequenced with a coverage ≥10× was evaluated for tumor samples.

The Burrows–Wheeler Aligner (BWA-MEM algorithm) was used for read mapping to the human reference genome (build hs37d5, based on NCBI GRCh37). PCR duplicates were discarded using the Mark Duplicates tool (Picard, Broad Institute, Cambridge, MA, USA), and then indel realignment and base quality score recalibration were performed with the Genome Analysis Toolkit (GATK, Broad Institute, Cambridge, MA, USA).

### 4.3. Mutational Profiling and Mutational Signature Analysis

In order to find somatic alterations that could explain the loss of expression pattern, MMR core genes (*MLH1*, *MSH2*, *MSH6*, *PMS2*), their associated genes *MSH3*, *MLH3,* and *PMS1*, and the additional *MUTYH*, *POLE,* and *POLD1* genes were considered to search for potentially pathogenic variants and predicting loss of heterozygosis (LOH) using somatic WES. LOH was defined as: (1) variant allele frequency (VAF) for a variant being > 80% higher than the average VAF of all tumor variants, and (2) corroborated by the analysis of shifts in expected VAF among germline polymorphisms within the same gene region. A putative LOH was defined as a VAF between 40–80% higher than the average somatic variant VAF [16]. In Appendix A are shown tumor reads depth and tumor reads from alternative allele and from reference allele.

Additionally, particular tumor features, including tumor mutational burden (TMB) and mutational signatures were analyzed. TMB was described as the total number of single nucleotide variants (SNVs) per megabase (Mb) accumulated in a given sample, assuming that an average WES sample accounts for 30 Mb with acceptable sequencing quality values. De novo mutational signature extraction was performed using SigProfilerExtractor computational framework, based on nonnegative matrix factorization [18]. Activities were calculated after the decomposition of the extracted signatures according to the reference set of mutational signatures described in COSMIC database (Mutational Signatures v3.1—Jun 2020). All data were integrated and visualized with Oviz-Bio platform (https://bio.oviz.org/ access date: 16 January 2021) [52].

### 4.4. Variant Calling and Filtering

The HaplotypeCaller (GATK), MuTect2 (GATK), and Strelka2 (Illumina) were used for SNV and indels calling for germline and tumor samples, respectively. Regarding variant annotation, several databases were considered, including SnpEff and dbNSFP for variant position and pathogenicity annotations. SIFT (prediction of damaging), PolyPhen2 (HumVar prediction of probably damaging or possibly damaging), and CADD (Phred score ≥ 20) were used for pathogenicity prediction of missense variants.

Germline WES data were analyzed using an in-house R language pipeline described in previous studies [53]. Functions related to CRC or cancer, in general, were prioritized. DNA repair functions were used as the main functional filter.

For tumor SNVs and indels, a similar filtering pipeline was used [54], restraining selected variants to those having a coverage ≥ 10 × both in germline and somatic samples, a tumor VAF ≥ 10%, and also selecting truncating (nonsense, splice site, and frameshift variants) or missense variants fulfilling at least two of the three missense pathogenicity tools criteria. (CADD, Polyphen2, and SIFT).

### 4.5. Germline Variant Prioritization and Validation

From the automatic filtering process performed for all variant types considered, a large number of potentially pathogenic alterations were identified for every sample. Thus, an additional prioritization process was implemented to select truly relevant alterations for the phenotype of interest. Using the access to both germline and somatic WES data, an integrated strategy based on Knudson’s two-hit hypothesis was developed to search for potential tumor suppressor genes (TSGs) associated with CRC germline predisposition. Genes with a potentially deleterious germline variant (first hit, SNV/indel) and a predicted second mutational event in the tumor (second hit, SNV/indel or LOH) were thus prioritized.

The prioritization process was completed with an additional stringent functional selection of the candidate genes compatible with the TSG model expected. Regarding function of candidate genes, DNA repair was the most stringent filter applied. Candidate genes identified in this process were manually curated according to functional evidence. Particular attention was paid to genes known to be involved in predisposition to CRC and other neoplasms by reviewing data present in OMIM (Online Mendelian Inheritance in Man; http://www.omim.org/ access date: 16 January 2021) and ClinVar (https://www.ncbi.nlm.nih.gov/clinvar/ access date: 16 January 2021). Final prioritized variants were validated by manual inspection of the WES data with Integrative Genomics Viewer [55].

## 5. Conclusions

In conclusion, we contribute new insights into the somatic characterization of LLS CRC, postulating *MSH3* and *POLD1* double somatic alterations as an underlying cause of MSI phenotype in CRCs with unexplained loss of IHC MSH2/MSH6 expression. Although the limited sample size of our study hinders the generalization of our observations and functional validation of our work is critical to provide translational conclusions, we propose intrinsic biological differences between LLS with and without somatic alterations that could facilitate targeted approaches for treatment, and we suggest new germline candidate genes in this scenario that need to be further investigated. We need to join efforts to understand this rare but increasingly common clinical scenario and develop a consensus on the terminology. For instance, should the term LLS include cases with double *MMR* gene somatic alterations, or not? Are these cases, especially early-onset or with a strong family history of cancer, truly “explained”? As Ladabaum pointed out [56], until we have a detailed molecular understanding of all LLS phenotypes, we need to guide intelligently our patients and their families in order to manage their future cancer risk, by making use of the clinical phenotype, our medical intuition, and common sense.

## Figures and Tables

**Figure 1 cancers-13-01259-f001:**
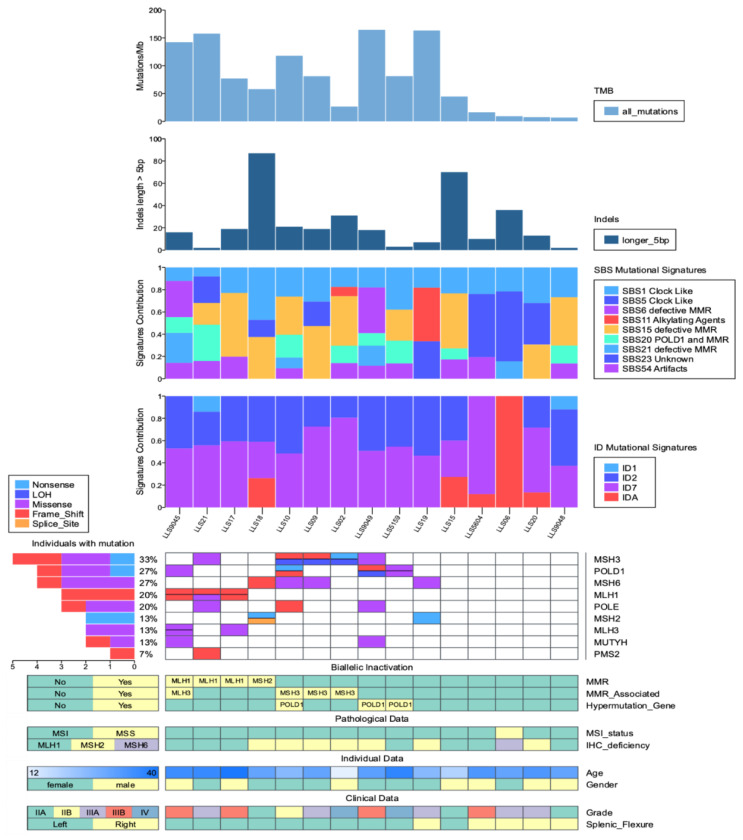
Genomic landscape of 15 early-onset colorectal cancer LLS patients. Samples are organized from left to right as MMR core genes biallelic inactivation, MMR-associated inactivation and hypermutated genes inactivation. Abbreviations TMB = tumor mutational burden (mutations/megabase); SBS = single base substitution; ID/Indels = short insertions and deletions; LOH = loss of heterozygosity.

**Table 1 cancers-13-01259-t001:** Clinical and-pathological features of the colorectal cancer (CRC) Lynch-like syndrome cohort.

Patient ID	Cohort	Age of Onset (Years)	Sex	Tumor Location	CRC Relation to Splenic Flexure	Stage	Histological Differentiation	Bethesda Clinical Criteria	MSI Status	MMR Protein Loss	Somatic Braf v600e	Biallelic Somatic Alteration Gene	TMB (Mutations/mb)
LLS02	Udaondo	12	male	rectum	Left	IV	well	1	MSI	MSH2/MSH6	WT	*MSH3*	26.7
LLS06	Udaondo	30	female	ascending colon	Right	IIIA	moderately	1;5	MSS	MSH6	WT	no alteration	9.4
LLS09	Udaondo	30	female	rectum	Left	IIIA	moderately	1;5	MSI	MSH2/MSH6	WT	*MSH3*	81.2
LLS10	Udaondo	25	female	descending colon	Left	IIB	moderately	1;3	MSI	MSH2/MSH6	WT	*MSH3*; *POLD1*	118
LLS15	Udaondo	19	male	descending colon	Left	IIA	no data	1;3	MSI	MLH1/PMS2	WT	no alteration	44.7
LLS17	Udaondo	40	male	descending colon	Left	IIIB	poorly	1;3	MSI	MLH1/PMS2	WT	*MLH1*	77
LLS18	Udaondo	30	female	descending colon	Left	IIA	moderately	1;3	MSI	MSH2/MSH6	WT	*MSH2*	57.8
**LLS19**	Udaondo	25	female	caecum	Right	IIIA	well	1;2;3	MSI	MSH2/MSH6	WT	**no alteration**	163.3
LLS20	Udaondo	33	male	ascending colon	Right	IIIA	moderately	1	MSI	MSH2/MSH6	WT	no alteration	7.6
LLS21	Udaondo	36	female	descending colon	Left	IIIA	moderately	1	MSI	MLH1/PMS2	WT	*MLH1*	157.7
LLS9049	Clinic	31	female	descending colon	Left	IIIB	moderately	1	MSI	MSH2/MSH6	WT	*POLD1*	164.6
LLS9045	Clinic	35	male	rectum	Left	IIIB	moderately	1	MSI	MLH1/PMS2	WT	*MLH1*	142.4
LLS5159	Clinic	37	female	sigma	Left	IV	poorly	1;3	MSI	MLH1/PMS2	WT	*POLD1*	81.4
LLS5604	Clinic	33	male	caecum	Right	IIIB	poorly	1;3	MSI	MLH1/PMS2	WT	no alteration	16.3
LLS9048	Clinic	30	male	ascending colon	Right	IIA	poorly	1;3	MSI	MLH1	WT	no alteration	6.8

MMR protein loss by IHC in FFPE slides. TMB as tumor mutational burden measured as all variants found in the WES analysis divided by 30 (mutations/MB). Histological differentiation: well (well differentiated adenocarcinoma), moderately (moderately differentiated adenocarcinoma), poorly (poorly differentiated adenocarcinoma). In bold, LLS19 has one single variant in MSH2 and also a very prevalent mutational signature related to a previous alkylating agent treatment.

**Table 2 cancers-13-01259-t002:** Somatic bi- and monoallelic variants in core and associated MMR genes.

Sample ID	Gene Name	HGVS.c	HGVS.p	Variant Impact	gnomAD	ClinVar	COSMIC	PolyPhen2	SIFT	CADD	Tumor AF	Average Sample AF	LOH AF Increase over Average (%)	LOH
**Biallelic Somatic Variants**														
**LLS02**	***MSH3***	**c.423C > A**	**p.Cys141 ***	**stop_gained**	**not present**	**not present**	**not present**	**-**	**-**	**38**	**0.60**	**0.35**	**73**	**yes**
**LLS09**	***MSH3***	**c.1148delA**	**p.Lys383fs**	**frameshift_variant**	**1.64399 × 10^−5^**	**Pathogenic**	**COSM1438888**	**-**	**-**	**-**	**0.46**	**0.33**	**40**	**yes**
**LLS10**	***MSH3***	**c.1148delA**	**p.Lys383fs**	**frameshift_variant**	**1.64399 × 10^−5^**	**Pathogenic**	**COSM1438888**	**-**	**-**	**-**	**0.61**	**0.35**	**75**	**yes**
LLS10	*POLD1*	c.583C > T	p.Arg195 *	stop_gained	0.000647	Conflicting	not present	-	-	37	0.36	0.35	5	no
LLS10	*POLD1*	c.2959delG	p.Asp987fs	frameshift_variant	0.000057209	not present	COSM3686158	-	-	-	0.40	0.35	15	no
LLS17	*MLH1*	c.129dupA	p.Ser44fs	frameshift_variant	not present	Pathogenic	not present	-	-	-	0.22	0.25	−14	no
LLS17	*MLH1*	c.1831delA	p.Ile611fs	frameshift_variant	not present	not present	not present	-	-	-	0.33	0.25	31	no
LLS18	*MSH2*	c.2251G > T	p.Gly751 *	stop_gained	not present	not present	not present	-	-	48	0.15	0.17	−11	no
LLS18	*MSH2*	c.2634+1G > A	-	splice_variant	not present	Likely_pathogenic	not present	-	-	27.6	0.19	0.17	14	no
LLS21	*MLH1*	c.199G > A	p.Gly67Arg	missense_variant	not present	Pathogenic	COSM1422567	D	D	34	0.13	0.18	−25	no
LLS21	*MLH1*	c.602delT	p.Val201fs	frameshift_variant	not present	not present	not present	-	-	-	0.24	0.18	36	no
LLS5159	*POLD1*	c.1562G > A	p.Arg521Gln	missense_variant	0.000126296	VUS	not present	P	D	31	0.22	022	4	no
LLS5159	*POLD1*	c.3047G > A	p.Arg1016His	missense_variant	not present	VUS	COSM7587416	P	D	29.6	0.27	0.22	25	no
LLS9045	*MLH3*	c.3694C > T	p.Arg1232Cys	missense_variant	3.24934 × 10^−5^	VUS	not present	D	D	31	0.28	0.28	4	no
**LLS9045**	***MLH3***	**c.1924T > C**	**p.Phe642Leu**	**missense_variant**	**not present**	**not present**	**not present**	**B**	**D**	**20.1**	**0.46**	**0.28**	**68**	**yes**
LLS9045	*MLH1*	c.588delA	p.Lys196fs	frameshift_variant	4.06276 × 10^−6^	Pathogenic	not present	-	-	-	0.22	0.28	−21	no
LLS9045	*MLH1*	c.1489delC	p.Arg497fs	frameshift_variant	4.06078 × 10^−6^	not present	COSM1422596	-	-	-	0.21	0.28	−23	no
**LLS9049**	***POLD1***	**c.2959delG**	**p.Asp987fs**	**frameshift_variant**	**0.000057209**	**not present**	**COSM3686158**	**-**	**-**	**-**	**0.31**	**0.19**	**57**	**yes**
**Monoallelic Somatic Variants**														
LLS09	*MSH6*	c.3552G > A	p.Met1184Ile	missense_variant	4.06835 × 10^−6^	not present	not present	B	T	28	0.30	0.33	−10	no
LLS10	*MSH6*	c.2875C > T	p.Arg959Cys	missense_variant	1.22491 × 10^−5^	VUS	not present	P	D	27.2	0.34	0.35	−1	no
LLS10	*POLE*	c.2091delC	p.Leu698fs	frameshift_variant	8.20836 × 10^−6^	VUS	COSM4612998	-	-	-	0.27	0.35	−21	no
LLS18	*MSH6*	c.3261dupC	p.Phe1088fs	frameshift_variant	0.00003449	Pathogenic	COSM13394	-	-	-	0.13	0.17	−23	no
LLS19	*MSH2*	c.1225C > T	p.Gln409 *	stop_gained	not present	Pathogenic	COSM7508782	-	-	41	0.16	0.15	7	no
LLS19	*MSH6*	c.1993G > A	p.Glu665Lys	missense_variant	4.06792 × 10^−6^	not present	not present	P	T	24.9	0.12	0.15	−18	no
LLS21	*MSH3*	c.3356T > C	p.Leu1119Pro	missense_variant	not present	not present	not prensent	D	D	27.3	0.14	0.18	−20	no
LLS21	*PMS2*	c.1239delA	p.Asp414fs	frameshift_variant	not present	not present	COSM150905	-	-	-	0.20	0.18	12	no
LLS21	*POLE*	c.1060A > G	p.Thr354Ala	missense_variant	not present	not present	not present	B	T	22.5	0.20	0.18	11	no
LLS9045	*MUTYH*	c.724C > T	p.Arg242Cys	missense_variant	5.29614 × 10^−5^	Pathogenic	COSM6954579	D	D	29.5	0.36	0.28	32	no
LLS9045	*POLD1*	c.735G > T	p.Glu245Asp	missense_variant	not present	not present	not present	D	T	22.9	0.27	0.28	−2	no
LLS9049	*MUTYH*	c.544C > T	p.Arg182Trp	missense_variant	1.21817 × 10^−5^	not present	COSM6922477	D	T	23.3	0.13	0.19	−32	no
LLS9049	*MSH3*	c.433G > T	p.Ala145Ser	missense_variant	not present	VUS	not present	P	T	17.64	0.15	0.19	−23	no
LLS9049	*POLE*	c.3176G > A	p.Arg1059His	missense_variant	1.21838 × 10^−5^	not present	COSM6965827	D	D	35	0.20	0.19	2	no

Rows in bold are showing when LOH is associated. GnomAD: allele frequency from gnomAD database. PolyPhen2: B: benign; P: probably pathogenic; D: deleterious; SIFT: T: tolerable; D: deleterious. Tumor AF: alternative allele frequency. Average AF: average of all exome variants over 0.1 of AF. LOH loss of heterozygosity. CADD: Phred score version 1.6. VUS: variant of uncertain significance. HGVS.c: coding variant. HGVS.p: protein level variant. * stop gained codon.

**Table 3 cancers-13-01259-t003:** Germline candidate variants. Results of the whole exome sequencing analysis with candidate variants with DNA repair functions.

ID	Germline Candidate Gene	HGVS.c	HGVS.p	Coding Impact	gnomAD	CADD	ClinVar	ACMG Classification
**Patients without biallelic inactivation in MMR core or their associated genes**								
LLS06	*ERCC6*	c.1670G > A	R557H (p.Arg557His)	missense	0.00018	27.6	Not present	VUS(PM1;PM2;PP3;BP1)
LLS20	*POLE*	c.1847G > A	R616H (p.Arg616His)	missense	0.0000358	26.4	Not present	VUS (PM2;PP3;BP1)
LLS5604	*EXO5*	c.23A > G	E8G (p.Glu8Gly)	missense	Not present	25.3	Not present	VUS (PM2;BP4)
LLS9048	*RECQL4*	c.1878+1G > C	-	donor_splice_site	Not present	32	Not present	Pathogenic (PVS1;PM2;PP3)
LLS15	*RAD52*	c.154A > T	I52L (p.Ile52Leu)	missense	Not present	25.1	Not present	VUS (PM2)
**Patients with biallelic inactivation in MMR core or their associated genes**								
LLS18	*PALB2*	c.2606C > T	S869F (p.Ser869Phe)	missense	Not present	29.2	VUS	VUS (PM1;PM2;PP3)
LLS21	*UVRAG*	c.937C > G	Q313E (p.Gln313Glu)	missense	Not present	26.3	Not present	VUS (PM2)
LLS09	*RAD54L*	c.767-2A > G	-	splice_acceptor	2.38663 × 10^−5^	33	Not present	VUS (PVS1;PP3)
LLS10	*RAD54L*	c.17C > T	A6V (p.Ala6Val)	missense	3.98108 × 10^−6^	32	Not present	VUS (PM2;PP3)
LLS02	*MCM2*	c.364C > T	R122W (p.Arg122Trp)	missense	1.66295 × 10^−5^	25.2	Not present	VUS (PM2;PP3;BP1)

ACMG classification: VUS: Variant of Uncertain Significance; PVS1: Pathogenic Very Strong 1; PM1: Pathogenic Moderate 1; PM2: Pathogenic Moderate 2; PP3: Pathogenic Supporting 3; BP1: Benign Supporting 1; BP4: Benign Supporting 4.

## Data Availability

Raw data are available upon request.

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
