# Peer review of "Comprehensive Genomic Characterization of Fifteen Early-Onset Lynch-Like Syndrome Colorectal Cancers"

_cancers, 2021, doi:10.3390/cancers13061259_

Round 1
Reviewer 1 Report
The authors have responded to each of my remarks, but apparently decided that their text did not need any changes. However, more readers may stumble over these issues and hence fail to appropriately appreciate the data described in this manuscript. I would therefore invite the authors to carefully look where in their manuscript questions may arise that need attention beforehand, even if they cannot definitely be answered.
- There are two questions here: first, can the mutational signature in MSH3 mutant tumors be explained by MSH3 defects or are they related to MSH2/6 defects? If I understand the paper correctly, the mutational signature in MSH3 defective tumors is not restricted to large repeat motifs, but includes single- and dinucleotide indels, and the TMB is high, which both correspond to MSH2/6 defects and not to MSH3 defects. Second, if MSH3 inactivation has an effect on MSH2/MSH6 activity, one would expect the opposite of what the authors propose. Loss of MSH3 would increase rather than decrease the level of MSH2/MSH6 complex.
So, I would invite the authors to explicitly discuss the mutation signature in MSH3 mutant tumors. They refer to EMAST, but it seems that the MSI seen in the MSH3 mutant tumors is restricted to single and dinucleotide slippage events. Also the mouse data they refer to do not provide evidence that single and dinucleotide slippage events can result from MSH3 inactivation at high frequency. So: describe exactly the mutational signature and explain the reader why this signature may relate to MSH3 deficiency.
- Allele frequency may indicate which mutations occurred first. So, stress this point in the text if you feel this is an important finding. However, I doubt whether this is the case. The AF is also used as indicator of LOH. E.g., in LLS10, the AF for MSH3 is 0.61; for POLD1 0.36. The authors argue from this that the MSH3 mutation occurred before the POLD1 mutation. But if the MSH3 AF is so high because of LOH, then this argument can no longer be made.
In conclusion, while I highly appreciate this manuscript, the findings related to bi-allelic MSH3 mutations need to my taste more attention in the Discussion section.
Author Response
We thank the reviewers for their insighful comments, which are helpin to improve our manuscript. We are answering below in red italics and highligting changes in the submitted version of our article.
Reviewer 1
The authors have responded to each of my remarks, but apparently decided that their text did not need any changes. However, more readers may stumble over these issues and hence fail to appropriately appreciate the data described in this manuscript. I would therefore invite the authors to carefully look where in their manuscript questions may arise that need attention beforehand, even if they cannot definitely be answered.
- There are two questions here: first, can the mutational signature in MSH3 mutant tumors be explained by MSH3 defects or are they related to MSH2/6 defects? If I understand the paper correctly, the mutational signature in MSH3 defective tumors is not restricted to large repeat motifs, but includes single- and dinucleotide indels, and the TMB is high, which both correspond to MSH2/6 defects and not to MSH3 defects.
We agree with the reviewer. The mutational signature of the MSH3 mutant tumors cannot be explained by MSH3 defects itself. The mutational signatures seen in these tumors also corresponds to the MMR deficiency. In fact, tumors in patients LLS03, LLS09 and LLS10 were MMR-deficient by IHC (MSH2-/MSH6-) and microsatellite unstable. Further, tumors in patients LLS09 and LLS10 also harbored a variant in MSH6 (with weak evidence of pathogenicity). Of note, most of MSI tumors also present EMAST.
Accordingly, we have added the following paragraph in the Discussion section:
“In our study, we found three patients that showed somatic MSH3 truncating variants associated with LOH, in MSH2-/MSH6- tumors. Although we hypothesize that the MSH3 alterations could be the underlying cause of the MMR inactivation, we cannot fully discard the contrary. Indeed, the mutational signatures associated with these tumors are mostly related to a MMR deficiency. We did not search for EMAST features and, therefore, we cannot differentiate between mono/dinucleotides alterations (associated to MMR deficiency) or polynucleotides alterations (associated to EMAST). However, it is well known that MMR-deficient tumors also shared an elevated microsatellite alterations at selected tetranucleotide repeats (EMAST) in previous studies [31]. On the other hand, no somatic mutations or LOH were found in MSH2 in these patients, and only two MSH6 VUS variants were detected (in LLS09 and LLS10).”
Second, if MSH3 inactivation has an effect on MSH2/MSH6 activity, one would expect the opposite of what the authors propose. Loss of MSH3 would increase rather than decrease the level of MSH2/MSH6 complex.
Again, we agree with the reviewer that the MSH3 inactivation could have an effect on MSH2/MSH6 activity. Based on our results by IHC, we observe an absence of expression of MSH2/MSH6. Our hypothesis does not take into account a possible imbalance in the protein levels of MSH3 and the MSH2/MSH6 complex, but rather considers that the absence of MSH3 due to the somatic defects detected could trigger the accumulation of DNA alterations, affecting MSH2/MSH6 among others and causing its absence of protein expression in the tumors of these patients. As previously mentioned, we cannot discard that it could be the other way around and the absence of MSH2/MSH6 due to somatic defects could be the initial event triggering the accumulation of DNA alterations including those in MSH3. We tend to favour the first option since the somatic alterations are more clearly detected in MSH3. As previosly commented, we are stressing this matter in the discussion with the added previous paragraph.
So, I would invite the authors to explicitly discuss the mutation signature in MSH3 mutant tumors. They refer to EMAST, but it seems that the MSI seen in the MSH3 mutant tumors is restricted to single and dinucleotide slippage events. Also the mouse data they refer to do not provide evidence that single and dinucleotide slippage events can result from MSH3 inactivation at high frequency. So: describe exactly the mutational signature and explain the reader why this signature may relate to MSH3 deficiency.
We thank the reviewer for this comment. To our knowledge, a specific mutational signature related to MSH3 deficiency is not yet known, besides the previous evidence that MSH3 deficiency causes EMAST or EMAST with low levels of MSI at loci with dinucleotide repeats in CRC. The COSMIC signatures used in our study do not differentiate between mono or polynucleotide repeats. However, it is likely that the MSH3 mutational signature shares some features with those currently known in the COSMIC database to be associated with MMR defects. Since our results favour this hypothesis, we are adding the sentence below in the Discussion section:
“To our knowledge, a specific mutational signature related to MSH3 deficiency is not yet known, besides the previous evidence that MSH3 deficiency causes EMAST or EMAST with low levels of MSI at loci with dinucleotide repeats in CRC. The COSMIC signatures used in our study do not differentiate between mono or polynucleotide repeats. However, it is likely that the MSH3 mutational signature shares some features with those currently known in the COSMIC database to be associated with MMR defects.”
- Allele frequency may indicate which mutations occurred first. So, stress this point in the text if you feel this is an important finding. However, I doubt whether this is the case. The AF is also used as indicator of LOH. E.g., in LLS10, the AF for MSH3 is 0.61; for POLD1 0.36. The authors argue from this that the MSH3 mutation occurred before the POLD1 mutation. But if the MSH3 AF is so high because of LOH, then this argument can no longer be made.
We agree with the reviewer that this point shoud be cautiously mentioned in the text. Accordingly, we have added the following sentence in the Discussion section:
“LLS09 and LLS10 showed concomitant MSH3/MSH6 and MSH3/POLD1 biallelic alterations, respectively. The LOH observed in MSH3 argues in favour that they occurred prior to those in MSH6 and POLD1. However, this matter should be taken very cautiously and further investigated in order to draw a more solid conclusion.”
In conclusion, while I highly appreciate this manuscript, the findings related to bi-allelic MSH3 mutations need to my taste more attention in the Discussion section.
In agreement with the reviewer, we have made several modifications in the Discussion section as previously stated.
Reviewer 2 Report
This study potentially identified MSH3 and POLD1 double somatic alterations as an underlying cause of a microsatellite instability (MSI) phenotype by the analysis of whole exome sequencing of patients. My major concern is that no validation in additional assay like site-directed mutagenesis, weakened the results, the mutation will be challenged, and the results may need further investigation.
Author Response
Reviewer 2
This study potentially identified MSH3 and POLD1 double somatic alterations as an underlying cause of a microsatellite instability (MSI) phenotype by the analysis of whole exome sequencing of patients. My major concern is that no validation in additional assay like site-directed mutagenesis, weakened the results, the mutation will be challenged, and the results may need further investigation.
We agree with the reviewer on these recommendations. Unfortunately, the suggested additional functional experiments could not be performed as part of the present study. Functional validation is indeed critical to provide translational conclusions. We believe it should be pursued if additonal studies in LLS patients further support our seminal findings. Accordingly, we have modified the following sentence in the Conclusion section to reinforce the need of a functional validation before concluding a role for MSH3 and POLD1 double somatic alterations in this clinical entity:
“Although the limited sample size of our study hinders the generalization of our observations and functional validation of our work is critical to provide translational conclusions, we propose intrinsic biological differences between LLS with and without somatic alterations that could facilitate targeted approaches for treatment, and we suggest new germline candidate genes in this scenario that need to be further investigated.”
Reviewer 3 Report
The authors have acknowledged the suggestion from the reviewer (first round) about the use of "variant" instead of mutation. However, the suggestion was:
The term “variant” should replace the term “mutation” or “polymorphism” WITH the following modifiers: (i) pathogenic, (ii) likely pathogenic, (iii) uncertain significance, (iv) likely benign, or (v) benign.
The authors should properly revise the manuscript using the correct terms.
Author Response
We thank the reviewers for their insighful comments, which are helpin to improve our manuscript. We are answering below in italics and highligting changes in the submitted version of our article.
Reviewer 3
The authors have acknowledged the suggestion from the reviewer (first round) about the use of "variant" instead of mutation. However, the suggestion was:
The term “variant” should replace the term “mutation” or “polymorphism” WITH the following modifiers (i) pathogenic, (ii) likely pathogenic, (iii) uncertain significance, (iv) likely benign, or (v) benign.
The authors should properly revise the manuscript using the correct terms.
We thank the reviewer for this comment. The term “variant” has replaced by the term “mutation” or “polymorphism” in the entire manuscript. Additonally, although all the variants mentioned in our manuscript are already classified in the Tables 2 and 3, we have reviewed the manuscript and incorporated these terms. Moreover, a new Supplementary Table 5 has been generated, where all somatic variants are classified following the mentioned ACMG criteria.
Round 2
Reviewer 2 Report
The revision improved the quality of manuscript, the results were supported by the experiments, and it is appropriate to be published in Cancers.
Author Response
Reviewer 2 Comments:
The revision improved the quality of manuscript, the results were supported by the experiments, and it is appropriate to be published in Cancers.
Response to Reviewer 2:
We thank the reviewer for his valuable comments on the manuscript. His comments has helped significantly to improve the quality of our work.
Reviewer 3 Report
Thanks for the revision of the term "variants" on the manuscript. Still some texts on the manuscript that are missing the correct term "pathogenic or deleterious variants".
Example: Introduction:..."It is an autosomal dominant condition caused by (pathogenic or deleterious) germline variants in a DNA mismatch repair (MMR) gene .."
Author Response
Reviewer 3 Comments:
Thanks for the revision of the term "variants" on the manuscript. Still some texts on the manuscript that are missing the correct term "pathogenic or deleterious variants".
Example: Introduction:..."It is an autosomal dominant condition caused by (pathogenic or deleterious) germline variants in a DNA mismatch repair (MMR) gene .."
Response Reviewer 3
We thank the reviewer for his valuable comments on the manuscript, that have really helped to improve the quality of our work. We have been revised the entire manuscript and changed all suggestions related to germline variants pathogenicity classification.
This manuscript is a resubmission of an earlier submission. The following is a list of the peer review reports and author responses from that submission.
Round 1
Reviewer 1 Report
Lynch-like syndrome is an increasing clinical challenge. Golobicki and colleagues have analyzed 15 LLS cases that were diagnosed < 40 y of age, and virtually all having an MSI tumor with immunohistochemical loss of MSH2/MSH6 or MLH1/PMS2. However, no germline MMR gene variant was detected. They found the following:
4/15 patients can be explained by the presence of two somatic deleterious , one case affecting MSH2 and three cases MLH1. One of the latter also had an MLH3 mutation plus LOH.
3/15 had a somatic MSH3 truncation plus LOH and IHC loss of MSH2/MSH6. One of these also had biallelic POLD1 mutations.
2/15 had biallelic POLD1 mutations and IHC loss of MSH2/MSH6 or MLH1/PMS2.
6/15 cases remained unexplained, but one of these showed monoallelic MSH2 and MSH6 variants without LOH and IHC loss of MSH2.
To further establish the role of MSH3 mutations, the authors studied 3,806 MSI CRC cases for MSH3 mutations. Although 67 cases were found with MSH3 mutations, 13 had at least one pathogenic MSH3 mutation but no mutations in the other MMR genes or MLH1 promoter silencing and remarkably, most of these were MSI.
Taken together, the results presented are highly interesting, well presented and shed new light on LLS that may help its clinical management.
The authors may consider the following questions:
- The results concerning MSH3 strongly suggest that biallelic MSH3 mutation may underlie the appearance of MSI tumors. Remarkably though the tumors also show loss of MSH2/MSH6 expression, MSI and a medium to high mutational burden. MSI and TMB point towards full MMR deficiency, consistent with absent MSH2/MSH6 staining. So the question is: if somatic MSH3 inactivation plus LOH underlies tumor development, how would this genotype lead to MSH2/MSH6 inactivation?
- The monoallelic mutations in MSH6 and POLE in the cases with biallelic MSH3 (LLS 09 and 10) are G>A, C>T and delC. These mutations are unlikely to be caused by the MSH3 defect. Conversely, the MSH3 mutations in LLS02, 09, 10 are C>A, delA and delA, which are consistent with the loss of MSH2/MSH6 staining. So, what is the evidence/indication that MSH3 mutation was causative? Is it possible that MSH3 mutations were secondary to loss of MSH2/MSH6?
- As referred to by the authors, biallelic germline MSH3 mutations have been associated with CRC predisposition. So, perhaps the combination of enhanced frameshift mutagenesis (by MSH3 loss) and increased point mutagenesis may be causative for early onset CRC. Does biallelic MSH3 mutation correlate with a specific ID signature?
- The cases with monoallelic MMR gene mutation may be fully MMR deficient by several mechanisms: a cryptic extra-exonic mutation in the other allele; dominant negative activity of the mutant allele; combination with other mutant alleles in the same cell, such as MSH6/POLE in LLS10, MH2/MSH6 in LLS19; MSH3/PMS2 in LLS21. Could the authors speculate which mechanism may underlie MMR deficiency and MSI in each case?
In conclusion, the findings on LLS cases are intriguing and raise interesting new questions about the etiology of LLS and as such represent a considerable step forward.
Reviewer 2 Report
Lynch-like syndrome manifests microsatellite instability (MSI), which is difficult to differ it from lynch syndrome that characterized by germline inactivation in DNA mismatch repair genes. The possible reasons that lynch-like syndrome showed MSI without gene mutations include: (1) unknown gene mutations in the germline that can drive MSI, (2) germline mutations occur in DNA MMR genes that are not identified, (3) genetic process in tumors other than germline mutations associated with the gene mutation causes MSI. The authors investigated the germline and tumor whole exome sequencing of patients, and identified potential genes as an underlying cause of MSI in lynch-like syndrome. My concerns are listed:
- may use functional assay like site-directed mutagenesis to validate the effects of somatic allelic alterations in MMR on the expression of associated genes in cells. IHC staining may explain the association between somatic biallelic alterations in MMR core and MMR-associated genes, but not the direct evidence.
- one of the limitation of this paper only performed whole exome sequencing, witch make up only less that 2 percent of whole genome, whole exome sequencing may miss valuable intronic regulatory mutations.
Reviewer 3 Report
Comprehensive genomic characterization of 2 early-onset Lynch-like syndrome colorectal cancers
The manuscript aimed to provide an overview of a clinical challenge entity as LLS. However, the number of analyzed cases is limited to 15 and it could not provide a comprehensive representation of this entity. I would suggest to the authors to rephrase the title and include the limitations in the manuscript.
Lines 50-51 of the introduction should be clarified as LS is the most common hereditary cancer syndrome, affecting an estimated 1 in 300 individuals and have reported to have increased incidences of cancers in the colon, rectum, endometrium, ovaries, stomach, small bowel, bile duct, pancreas and upper urinary tract.
Lines 56: Would appreciate to clarify and add a reference about …”PMS2 and MSH6 are currently more prevalent on a population basis…”: Does it mean more prevalent than MLH1 and MSH2?
It would have been more appropriate to use the term “variant” throughout the manuscript, which is in line with the standards and guidelines for the interpretation of sequence variants [e.g. Richards et al, 2015]. The term “variant” should replace the term “mutation” or “polymorphism” with the following modifiers: (i) pathogenic, (ii) likely pathogenic, (iii) uncertain significance, (iv) likely benign, or (v) benign.
Results: Overall, the numbers are not well presented as there are contradictory results. For example: Table 1 described all the 15 cases, out of them, 5 did not have an alteration of a biallelic somatic. However, the section 2.2. had described not congruent numbers (lines 96-97: must be 2 cases with somatic pathogenic variants, including LLS19); line 110 it is not 6 remaining cases, there are 5.
It is also applied to section 2.3 where case LLS19 had reported to MSH2 somatic pathogenic variant.
When it comes to MMR pathogenicity classification , we suggest to follow the expert committee from InSIGHT.
Round 2
Reviewer 2 Report
Very interesting findings. Mainly focused on somatic variants in MMR and MMR-associated genes. But again functionally pindown and validate the actual causal variant(s) is critical to provide translational conclusions.
Reviewer 3 Report
The authors have not clarified the following comments:
- Prevalence of PMS2 and MSH6. The authors are stating it based on only one study. Should reconsider and revise the literature in this regards.
- Not possible to see the whole table 2.
- The authors have not included InSIGH classification on the methods.